# The Bearing Stiffness Effect on In-Wheel Motors

**Matej Biček [1], Raphaël Connes [1], Senad Omerović [1], Aydin Gündüz [2],**
**Robert Kunc [3],\* and Samo Zupan [3]**

[1]  Elaphe Ltd., Teslova 30, Ljubljana 1000, Slovenia; matej.bicek@elaphe-ev.com (M.B.);
raphael.connes@elaphe-ev.com (R.C.); senad.omerovic@elaphe-ev.com (S.O.)

[2]  Turkish Aerospace Inc., Ankara 06980, Turkey; dr.aydingunduz@gmail.com

[3]  Faculty of Mechanical Engineering, University of Ljubljana, Ljubljana 1000, Slovenia;
samo.zupan@fs.uni-lj.si

\*  Correspondence: robert.kunc@fs.uni-lj.si; Tel.: +386-1-477-1508

**Abstract:** In-wheel motors offer a promising solution for novel drivetrain architectures of future electric vehicles that could penetrate into the automotive industry by transferring the drive directly inside the wheels. The available literature mainly deals with the optimization of electromagnetically active parts; however, the mechanical design of electromagnetically passive parts that indirectly influence motor performance also require detailed analysis and extensive validation. To meet the optimal performance of an in-wheel motor, the mechanical design requires optimization of housing elements, thermal management, mechanical tolerancing and hub bearing selection. All of the mentioned factors have an indirect influence on the electromagnetic performance of the IWM and sustainability; therefore, the following paper identifies the hub bearing as a critical component for the in-wheel motor application. Acting loads are reviewed and their effect on component deformation is studied via analytically and numerically determined stiffness as well as later validated by measurements on the component and assembly level to ensure deformation envelope and functionality within a wide range of operations.

**Keywords:** Air-gap; hub bearing; in-wheel motor; mathematical stiffness model; validation tests

## 1. Introduction

Conventional mobility with Internal Combustion Engines (ICE) and complex drivetrains as shown on Figure 1 is facing competition with electric vehicles within the passenger car market and lately also within commercial segments. Showcasing electric vehicles is a thing of the past as they are pushed by incentives and national or associative directives to being frequently driven on global roads.

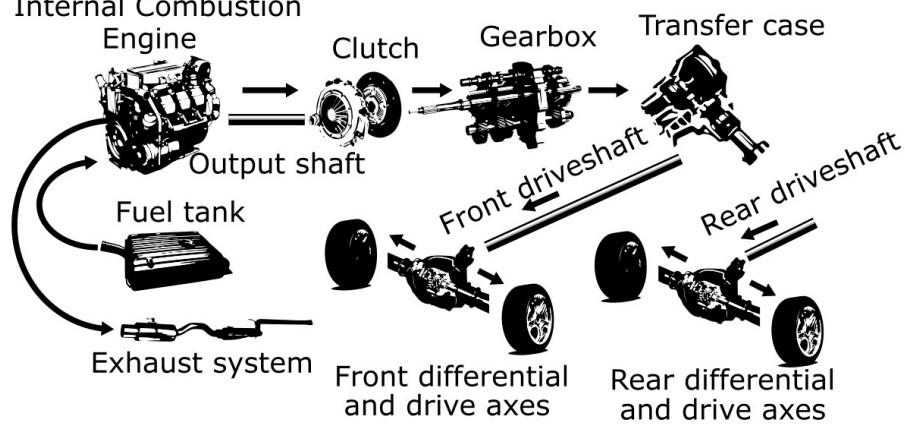

**Figure 1.** Internal Combustion Engines (ICE) drivetrain with an all-wheel drive with obvious complexity [1].

The automotive evolution includes charging infrastructure establishment, energy source development and propulsion architecture selection. In-Wheel Motor (IWM) platforms (Figure 2) intrinsically allow higher design space, lowering of the vehicle's center of gravity, reduction of required parts for vehicle propulsion [2,3], and consequentially offers cost reduction potential [4,5]. Higher energy efficiency and increased range [6–8] are also met as no mechanical transmission is required and the wheels are propelled directly [4,9,10]. More space for passengers and cargo [10–13] allows chassis designers to utterly change the way cars look and perform [14,15] with components not needing fixed mechanical coupling allowing free arrangement in the vehicle.

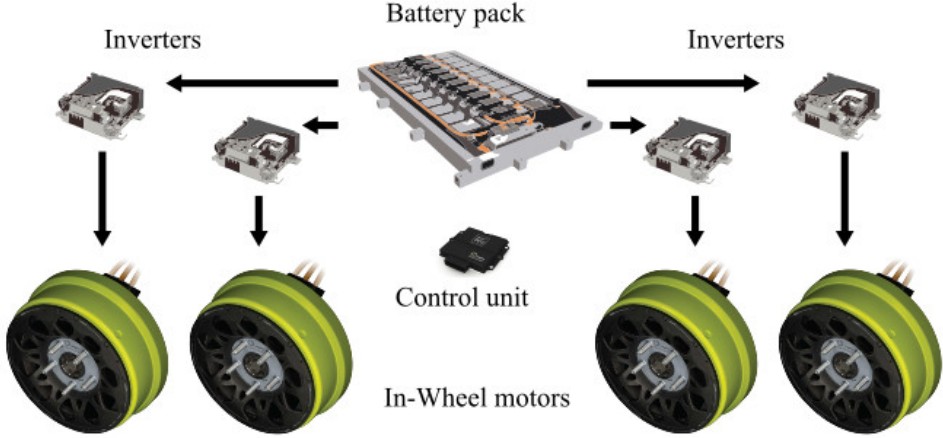

**Figure 2.** In-Wheel Motor (IWM) propulsion platform with Elaphe M700 all-wheel drive showing simplicity [1].

The reason for not having this technology already adopted in the automotive world likely stems from the lack of required know-how for the design of high torque density, innovations in high diameter sealing with low loss generation, noise, vibration, and harshness (NVH), lightweight design, with stiff and durable housing components. The most common Permanent Magnet Synchronous Motor–PMSM for in-wheel applications has an outer rotor with a relatively small air gap between rotor and stator [8]. Identification of mechanical failure modes and effects have been presented in a review paper [13] with keywords such as unsprung mass, eccentricity, moment of inertia, vibrations and hub bearing faults. The latter is rated as a component with one of the highest Risk Priority Numbers (RPN) and this paper describes in detail the reasoning behind it.

The following chapters deal with the identification of bearing function, loads acting on it, stiffness evaluation with numerical and experimental approach as well as validation on bearings and assembled IWM.

## 2. Bearing Function in IWM

Integration of a hub bearing unit into the in-wheel motor is objected to ensure the rotation whereas offering required stiffness to support axial and vertical loads as shown in Figure 3.

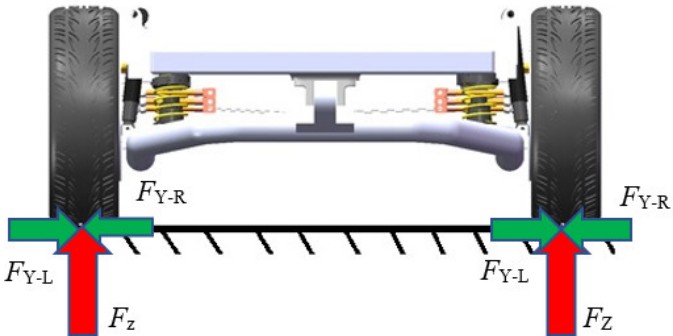

**Figure 3.** Axial and radial loads acting on rear wheels of a vehicle.

Vertical loads are directed opposite to gravitational force, whereas axial loads occur in the cornering direction. The bearing is most affected by the bending moment resulting from the lateral (cornering) force, defined as $M_{X-Y}$, which acts via pneumatic tire's effective rolling radius and rim, as shown in Figure 4. $F_Y$ is shown as an example for left cornering shown as $F_{Y-L}$ in Figure 3.

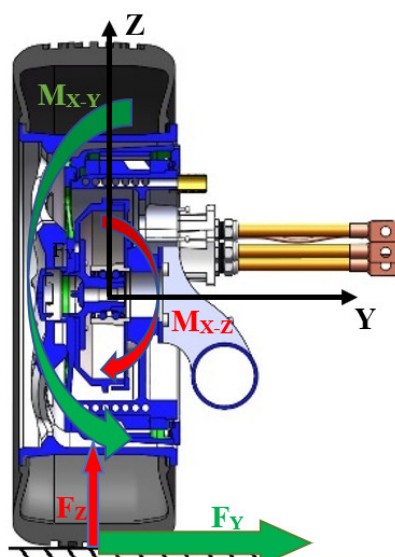

**Figure 4.** Bending moments generated by the axial and radial force.

Moment $M_{X-Z}$ is much smaller since it is resulting from the vertical force and the small distance from the tire center to the point of rotational deflection. As identified, the most critical loads are severe braking, cornering and driving over a road pothole/obstacle causing an impact load. $M_{X-Z}$ from vertical impacts or $M_{X-Y}$ from severe cornering can reach values that result in large deflection angles and should be anticipated during the design stage. Hub bearing deflection is less problematic for conventional vehicle corners, where the deflection acts on the movement of disc brake towards braking pads inside the caliper (Figure 5). The objective of every brake manufacturer is to design a braking assembly, which will be functional and not affect wheel rotation during severe cornering.

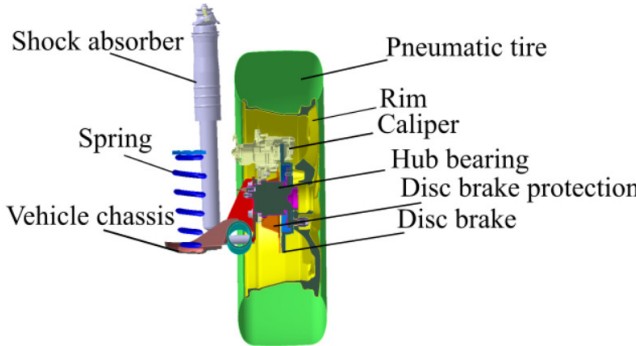

**Figure 5.** Conventional suspension with components in display. Hub bearing deflection angle *β* is limited by braking pads inside the caliper and can be larger in comparison to when integrated inside an IWM.

Bearing manufacturers for this reason develop different solutions to reduce the brake pedal travel and improve brake caliper piston knock-back [16]. In case of too large hub bearing deflection, brakes can also endure non-uniform wear of rotor and brake lining which has a negative effect on vehicle handling performance. The air gap between rotor and stator of the most common IWM layout is shown in Figure 6 as the envelope in which hub bearing deflection is allowed. The gap is also used to cover elastic deformations of motor housing due to accelerations, thermomechanical loads, production and assembly tolerances, deformations due to press fits and residual stresses resulting from manufacturing processes.

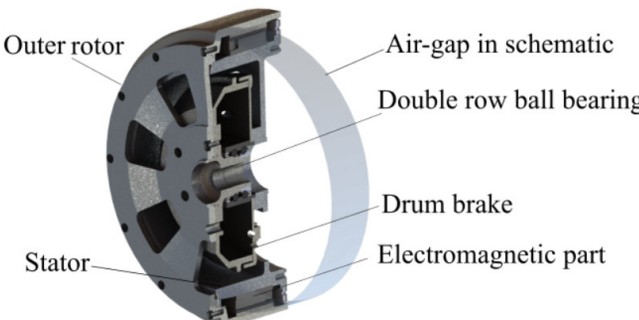

**Figure 6.** Elaphe M700 IWM with central hub bearing layout in section cut and air gap in schematic circumference.

The nominal size of the air gap for IWMs is normally designed to be 1 mm. In order to define a feasible value for the electromagnetic design, a comprehensive design of the mechanical parts is essential. Ensuring its size is therefore closely connected with electromagnetic performance of the motor and the mechanical torque as an output. One example of air-gap dimensional change in correlation with the output torque is presented in Figure 7, where the average output torque is calculated in relation to the generated magnetic flux density influenced by the air gap.

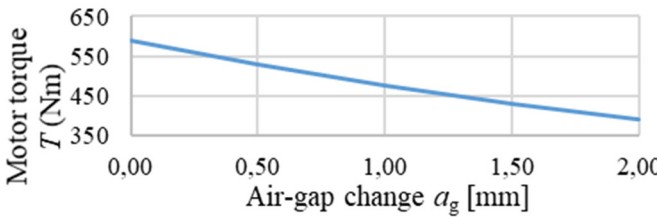

**Figure 7.** Torque output in relation to air-gap size for M700 IWM.

In addition to these limitations, large diameter seals also require stable operation and consequential movement of components within a specific range. For the presented case study, the

seal operation field is defined to be ± 0,2 mm. The presented methods for hub bearing stiffness identification are applicable and the so-called hub bearing deflection should be known upfront in the design stage in order to prevent contact between the static and rotating parts of the motor in the worst-case conditions and to fulfil the requirements for high-diameter sealing. In this study, the objective is directed to automotive generation 3 hub bearings as defined in [17] and shown in Figure 8.

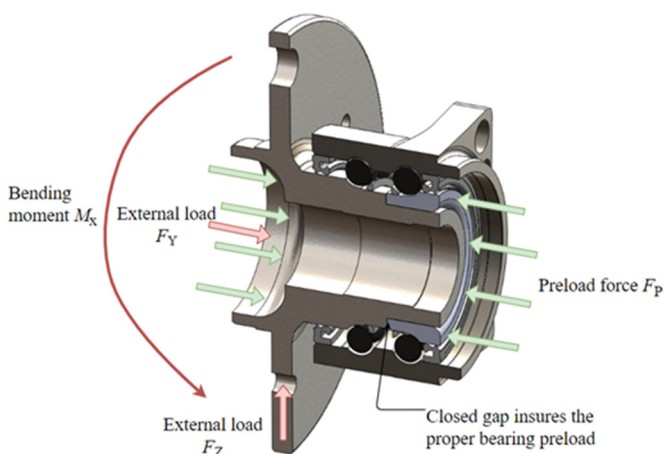

**Figure 8.** Hub bearing section cut with schematically applied bending moment $M_x$ resulting in deflection angle $\beta$ of rotational part in relation to static part.

Tolerance stack analysis of the complete IWM as shown in Figure 9 should include all contributing factors to define the maximum allowed deformation due to hub bearing deflection, and additional research should be made to understand the deflection mechanism to accordingly select or design a favorable hub bearing.

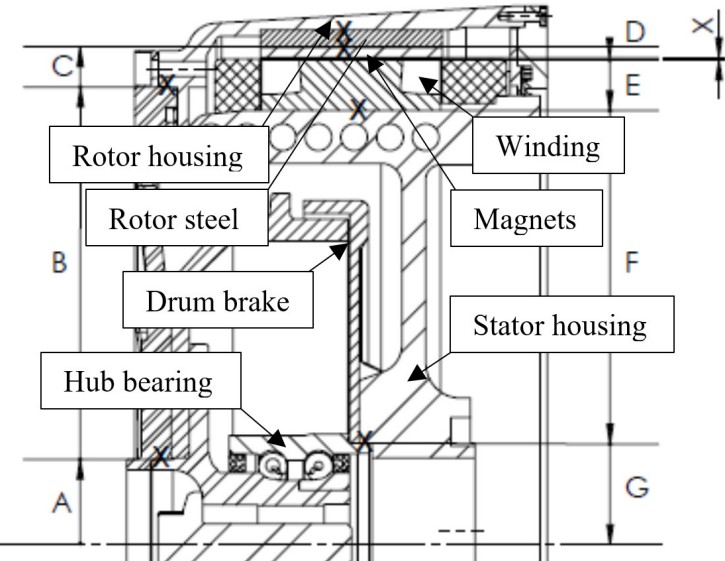

**Figure 9.** Radial tolerance stack (TS) path of an IWM with integrated drum brake in section cut. X markings are presenting press fits without effect on TS.

## 3. Understanding the Mechanism of Bearing Deflection

Double row angular contact ball bearings have a clear advantage regarding axial and radial stiffness in comparison to single row ball bearings. They can carry the axial loads in both directions whereas offering a substantial level of combined load carrying capacity where the radial, axial and moment loads act simultaneously. Back-to-back mounting additionally offers high moment rigidity,

which may be further improved by preloading. Due to these advantages, they are used in numerous applications besides automotive wheel-hubs [18]. Several authors have made investigations and developed different approaches for performing FEM simulations of bearing deflection in order to reduce the required time from design to prototyping. All reviews, including results, are noted below in Table 1.

**Table 1.** Obtainable research on hub bearing deflection angle done for conventional vehicles, used load cases and concluding deflections.

| Source | Applied load | Deflection angle $\beta$ – simulated [°] | Deflection angle $\beta$ – measured [°] |
|---|---|---|---|
| [19] | $F_x = 10$ kN | 0.48 | 0.49 |
| [20] | $a_x = 0.3$ g | 0.145 | 0.161 |
| [20] | $a_x = 0.6$ g | 0.229 | 0.231 |
| [21] | $M_x = 2$ kNm | 0.300 | 0.250 |
| [17] | $M_x = 3.3$ kNm | 0.620 | 0.710 |
| [22] | $M_x = 1.31$ kNm | 0.261 | 0.217 |

The stiffness concept is derived from the theory of elasticity as the relation between deformation of the element and input of external loads on the component by:

$$F = k\delta \tag{1}$$

with $F$ being the applied force, $k$ the stiffness of the component and $\delta$ the deformation. This relation between force and deformation is normally made in a matrix form by:

$$\begin{Bmatrix} F_1 \\ F_2 \end{Bmatrix} = \begin{bmatrix} k & -k \\ -k & k \end{bmatrix} \begin{Bmatrix} \delta_1 \\ \delta_2 \end{Bmatrix} \tag{2}$$

with $F_1$ and $F_2$ being the applied forces and $\delta_1$, $\delta_2$ the resulting deformations in nodes 1 and 2, respectively [23]. Stiffness matrix of a double row angular contact ball bearing is obviously more complex; however, many publications exist that differ in mathematical models and geometry of analyzed bearing layouts. Figure 10 schematically represents the existing approaches for the definition of the stiffness matrix for a double row ball bearing.

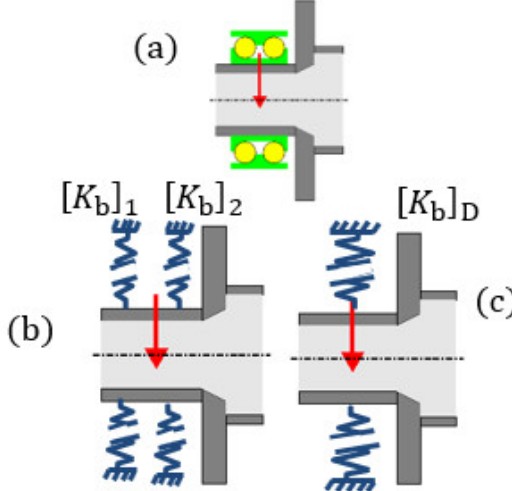

**Figure 10.** (**a**) Schematic view of an assembled double row bearing [24]; (**b**) model with two stiffness matrices; (**c**) model with one stiffness matrix. (**b**) and (**c**) with multi-dimensional non-linear springs (no torsional stiffness).

The major complexity comes from the nonlinear contact characteristics inside the bearing (between balls and raceways), which makes the bearing stiffness nonlinear and load-dependent. In the most recent publications bearing stiffness is defined with a 5 × 5 stiffness matrix (assuming the bearing is free to rotate around its axis, so torsional stiffness is disregarded), for a defined loading condition. For double row bearings (such as in Figure 10 (a)), the stiffness may either be identified by two matrices (one matrix for each row as shown in Figure 10 (b)), or one stiffness matrix for the double row unit (see Figure 10 (c)).

Although the two methods yield identical system deflections, superposing the moment stiffness elements of two single rows may not result in the moment stiffness of the double row unit. In this study, the latter approach is adopted, as the stiffness of the double row unit dictates the angular displacement.

To define the load-deflection characteristic of a single rolling contact, calculation from (1) is converted into:

$$Q_j = K_n \delta_j^n \tag{3}$$

where $Q_j$ represents the resulting normal load on a single rolling element at position j, $K_n$ is a stiffness constant accounting for geometry and material (also known as Hertzian stiffness constant or load-deflection factor), and $n$ is a value (exponent) defining the nature of the contact; for point contacts (i.e., ball bearings) $n = 1.5$ [18]. By adding the contribution from each rolling element, (3) can be translated into a complex relationship between the bearing load vector ($f_b$) and the bearing deflection vector ($q_b$) [25]. Bearing stiffness matrix can then be obtained by applying the mathematical definition of stiffness and taking partial derivatives of each load term against each deflection term.

Deflection (movements and twists) and load (forces and torques) vectors are described by Equations (4) and (5), respectively, whereas the stiffness matrix is shown by Equation (6), which included partial translational and rotational (around coordinate axis–elements with $\theta$ indexes) stiffness according to Cartesian coordinate system:

$$q_b = \left\{ \delta_x, \delta_y, \delta_z, \beta_x, \beta_y \right\}^T, \tag{4}$$

$$f_b = \left\{ F_x, F_y, F_z, M_x, M_y \right\}^T, \tag{5}$$

$$[\mathbf{K_b}]_D = \begin{bmatrix} k_{xx} & k_{xy} & k_{xz} & k_{x\theta_x} & k_{x\theta_y} \\ k_{yx} & k_{yy} & k_{yz} & k_{y\theta_x} & k_{y\theta_y} \\ k_{zx} & k_{zy} & k_{zz} & k_{z\theta_x} & k_{z\theta_y} \\ k_{\theta_x x} & k_{\theta_y x} & k_{\theta_z x} & k_{\theta_x \theta_x} & k_{\theta_x \theta_y} \\ k_{\theta_y x} & k_{\theta_y y} & k_{\theta_y z} & k_{\theta_y \theta_x} & k_{\theta_y \theta_y} \end{bmatrix}. \tag{6}$$

## 4. Loads on IWM

Two types of load cases are important for selecting the most suitable bearing for IWM wheel hub system. The first load case type is the one leading to the most damage in the bearing. It includes impacts such as driving over an obstacle or a pothole and repetitive loading. The second load case type is the one leading to the highest bending deflection resulting from severe cornering.

Loads can be prescribed by the application or derived from a test-mule vehicle. In the presented case study, a BMW X6 was converted with IWM propulsion architecture USING 4X Elaphe L1500 motors with specification shown in performed skid pad test described in ISO 4138-2012 [25].

The vehicle was driving in a circle of radius $r = 35.5$ m and the speed was increasing with every lap as shown below in Figure 11. Above $v = 70$ km/h $= 19,44$ m/s the vehicle started drifting sideways, which shows the limit friction of the vehicle. Since this paper is not about tire physics, it can be assumed that the maximal lateral acceleration is also the maximal tire friction by:

$$\mu = \frac{v^2}{r\,g} = 1.08 \tag{7}$$

with *v* being the vehicle speed, *r* turning radius and *g* gravitational acceleration.

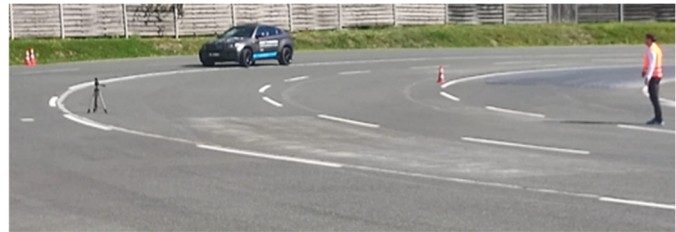

**Figure 11.** In-wheel motor vehicle based on BMW X6 during skid pad test at constant radius.

During conversion of the vehicle, the ICE, gear box, fuel tank, differential and transmission were removed (−565 kg) and 4 IWM motors, 4 inverters and a battery pack were added (+520 kg). This resulted in a modified Centre of Gravity (CoG) position of 131 mm more to the back and 72 mm lower than in the original state. Calculated lateral forces on front tires are presented below in Figure 12. Forces are applied from −2500 N at 7 m/s² (0.71 g) of lateral acceleration on inner front wheel to +13600 N at 11 m/s² (1.12 g) of lateral acceleration on external front wheel (Figure 13). The reduction of CoG height decreased the maximal load (linked to maximal IWM deflection) by 7.5%.

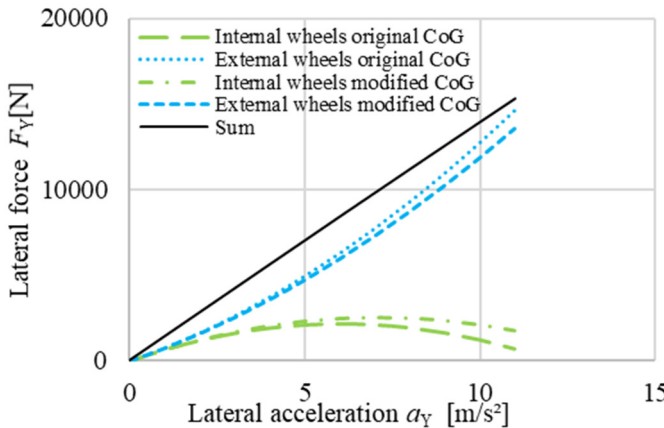

**Figure 12.** Lateral force linked with weight transfer on front wheels depending on the lateral acceleration.

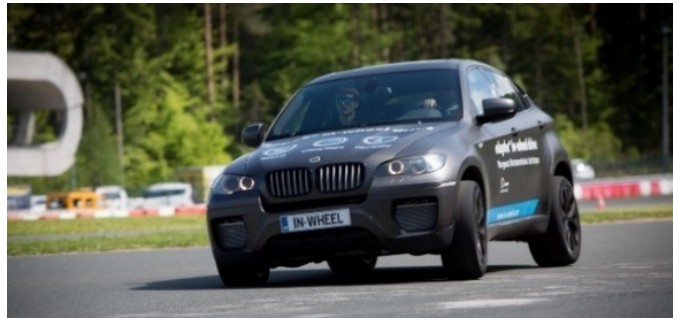

**Figure 13.** Photo from one of the test sessions, where accelerations were measured for severe cornering.

The loads are within 3% of the maximal loads calculated with multibody dynamics software when modelling double lane change (Figure 14). This provides useful information on the frequency of such extreme events.

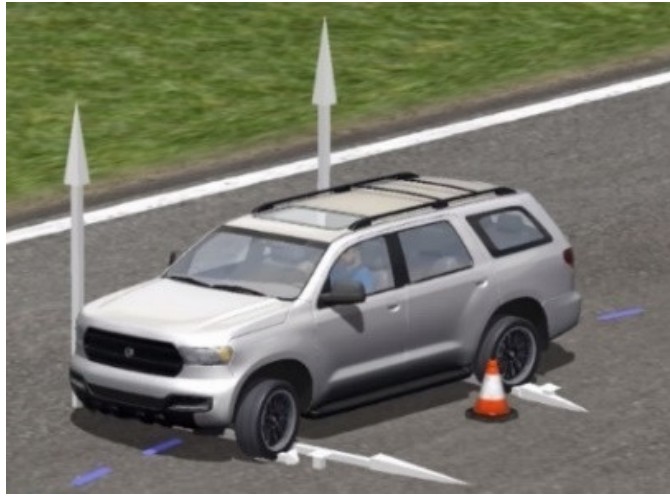

**Figure 14.** Simulated version for severe cornering shown in Figure 12.

When loads are applied in the middle of each tire contact patch, they are transferred to the bearing raceways center by:

$$M_X = M_{X-Y} + M_{X-Z} = F_Y R_w + F_Z a \qquad (8)$$

$$M_Z = F_X a \qquad (9)$$

with $M_X$ being the bending moment in longitudinal direction, $F_Y$ force in the axial direction, $R_w$ the tire radius, $F_Z$ the vertical force on the tire and a, the axial distance between the middle of the contact patch and the middle of both ball raceways of the bearing, $M_Z$ the vertical bending moment and $F_X$ the longitudinal force. The distances $R_w$ and $a$ are not fixed. During cornering the extra weight on outer wheels compress the tires so much that $M_{X-Y}$ decreases (–4% of $M_X$) and due to the axial deformation on the tire $M_{X-Z}$ increases (+8% of $M_X$).

Pothole and vibration-induced fatigue were considered as an unsprung mass system; thus, loading is replaced by acceleration. For pothole, the complete motor was subjected to two successive 100 g jolts in both vertical directions. To see if the bearing could sustain vibration, the acceleration-frequency power spectral density with random signal in accordance to ISO 16750 3:2007 was made as a validation test upfront [26].

## 5. Determination of Stiffness

The stiffness of the wheel hub systems is greatly determined by the hub bearing. Its materials and geometry are the two most important aspects. Obviously, larger bearings have higher stiffness terms, however not all bearing parameters influence the stiffness in the same way. In fact, the internal geometry (rolling elements, race ways) rather than basic external geometry (hub bearing housing) affects the bearing stiffness terms. For ball bearings, these parameters are the pitch diameter, ball diameter, number of balls in each row, contact angle, and radii of the inner and outer raceway curvatures (raceway conformities). Thus, it is important to know the internal design details of the bearing to accurately estimate the bearing stiffness.

Bearing mounting arrangement and row separation distance *D* (Figure 15) are two other important design parameters, as these parameters determine the effective spread *E* (Figure 15) together with the pitch diameter and contact angle. The effective spread is a measure of the moment stiffness of the bearing. Back-to-back or O-arrangement bearings have much larger spread than face-to-face X-arrangement bearings (Figure 15). Due to this reason, automotive hub bearings are always mounted back-to-back (O).

The moment stiffness of the hub bearing may be further improved by preloading. This is discussed in more detail in the following section.



**Figure 15.** Illustration of back-to-back (O) and face-to-face (X) bearing arrangement.

In order to evaluate the resulting effects of all these parameters combined, an analytical bearing design/analysis tool EBECA (Elaphe BEaring Calculator) has been developed (graphical interface shown in Figure 16). This software prompts for the bearing geometry and mounting information (mentioned above), along with the bearing preload and load vectors acting on the bearing. Due to nonlinearity, an iterative method based on the simplex algorithm is then used to output the bearing deflection vector, stiffness matrix and internal load and stress distributions.

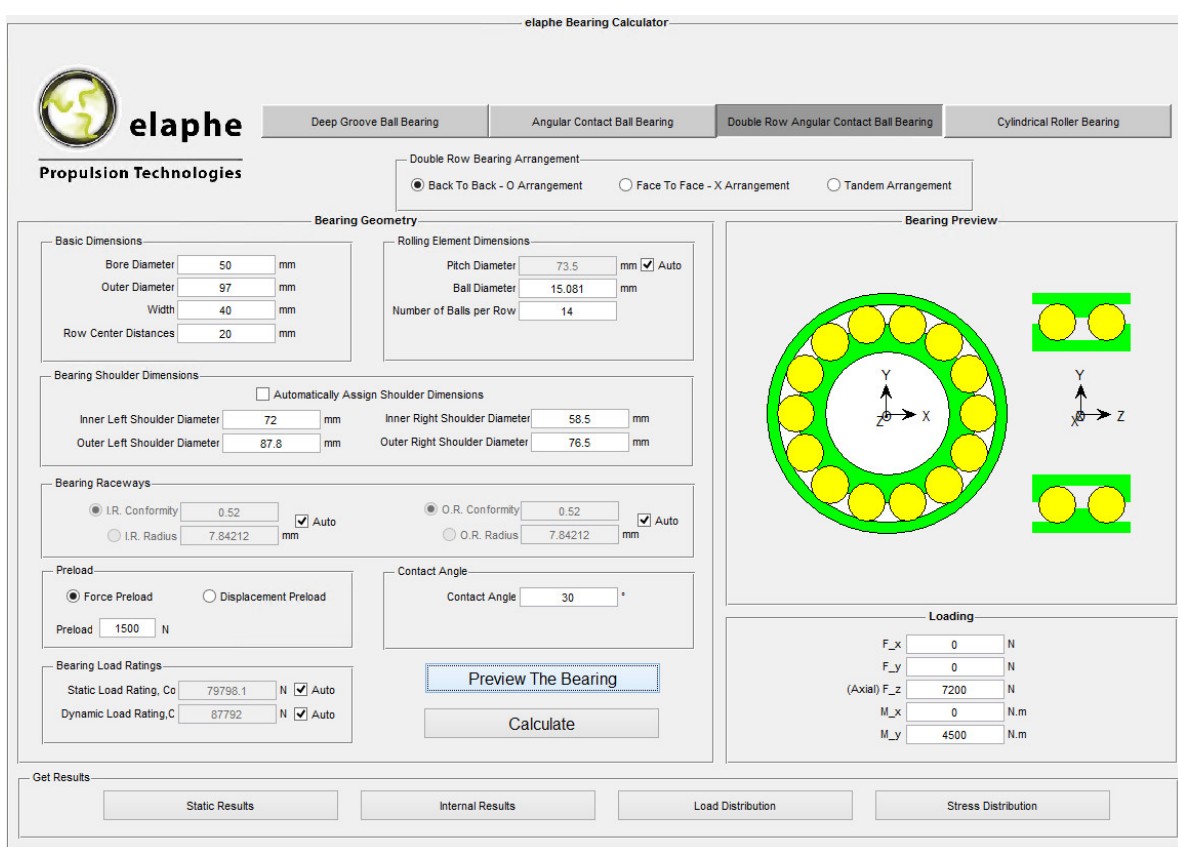

**Figure 16.** Input data for standard BMW X6 front bearing in the analytical bearing design/analysis tool.

In this example, the data of the BMW X6 front hub bearing were fed into the interface. Repeating this analysis for several operating points and for various bearings, the deflection results are obtained as shown in Figure 17. For benchmark purposes, several other conventional hub bearings have also been analyzed.

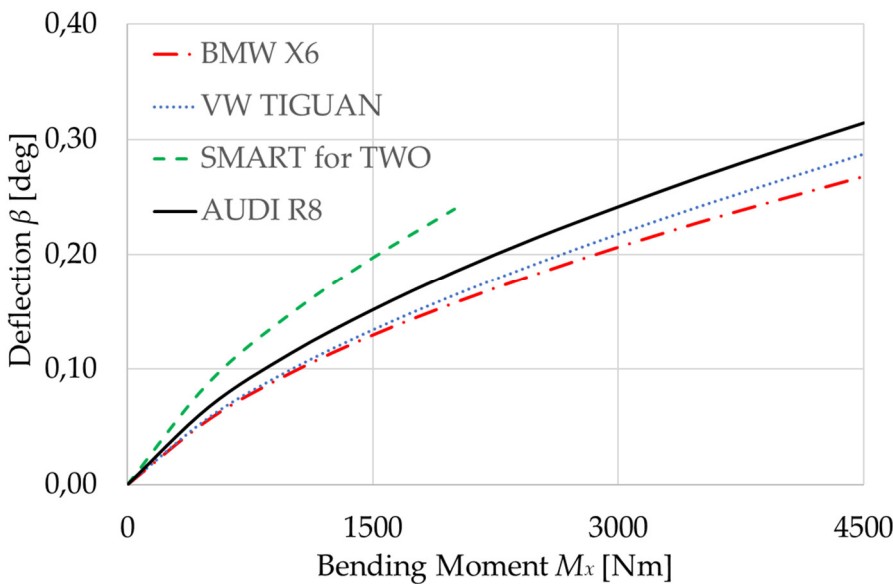

**Figure 17.** Analytically calculated bearing deflection without including the elastic deformation of bearing housing.

EBECA analytical bearing tool is used to determine load distribution and stresses within the bearing shown in Figure 18.

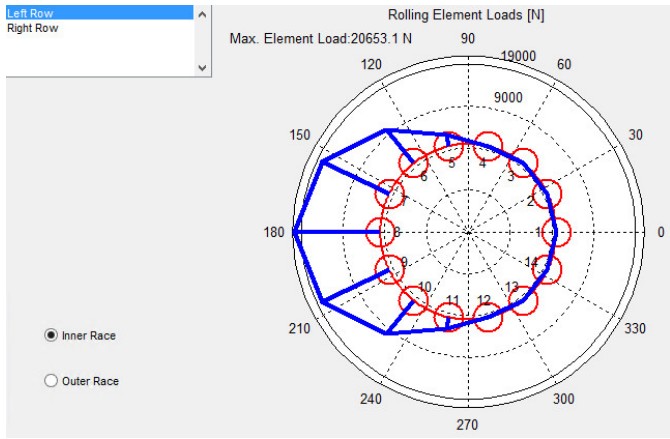

**Figure 18.** Load distribution calculated using bearing design/analysis tool.

To take into account static and dynamic flanges deformations the bearing geometry and load were simulated using Finite Element Methods (FEM). The deformation shown is not to be neglected, as it accounts for an overall contribution of 30% of the total angular deflection, as seen below in Figure 19. The analysis also confirms that the highest stresses near the bolt threads are below the element's Yield point. Elastic deformation of hub bearing housing elements has been numerically simulated for several load cases and added to the calculated bearing deflections for several hub bearings in the later sections.

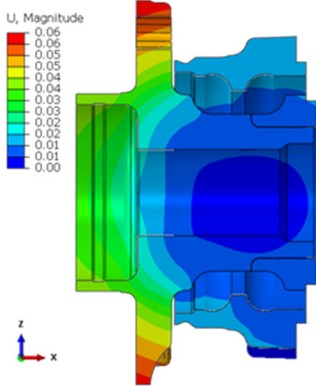

**Figure 19.** Visualization of results for a numerical simulation of BMW X6 M front bearing Y-plane deformation in [mm] with maximum bending moment applied around Y axis.

## 6. Validation of Hub Bearing Stiffness on Test Rig

In order to validate the bearing stiffness calculations, several bearing deflection tests were performed on a custom-made bearing deflection test rig (Figure 20). Bearing stiffness properties were derived from the bearing rotor flange displacements measured with laser triangulation and the load applied with hydraulic cylinders.

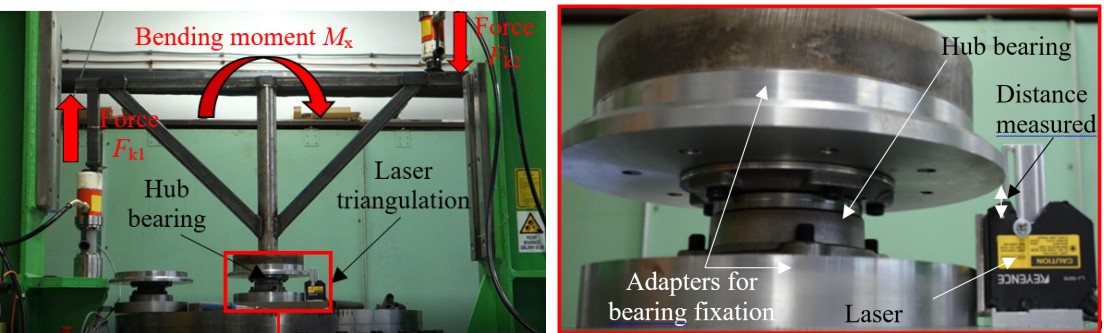

**Figure 20.** BMW X6 front bearing on the deflection test rig at the University of Ljubljana (UL), Faculty of Mechanical Engineering (FME) [27].

Tests were performed for three different load cases acting on the hub bearing, as described in Figure 21.

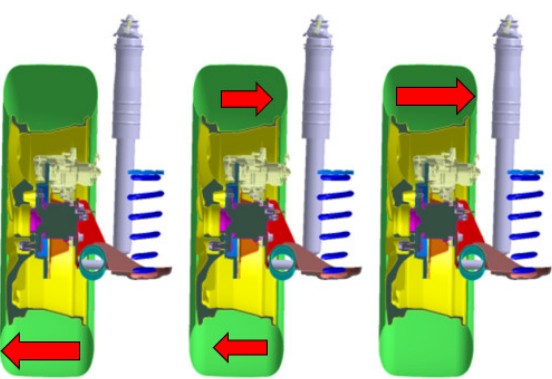

**Figure 21.** Applied load cases (LC) for performing deflection tests at FME.

For bearing performance, particularly in the range of bearing stiffness, the bearing preload plays a major role [28]. This preload on the bearing rolling elements is typically achieved by closing the gap or play between the two raceway rings. Depending on the bearing design, gap closing may be obtained during the manufacturing process or applied during the bearing installation. Sustaining a

sufficient bearing preload under all working conditions is critical and, in this case, ensured with sufficient axial prestress. This is ensured with a sufficient level of external clamping force. Influence of the amount of clamping force was evaluated with the variation of bearing tightening torque. This parameter is crucial when designing the IWM preload shaft used for generating the clamping force.

Bearing deflection measurements for the maximum bending moment are presented in Figure 22. The load was limited to 4500 Nm (except for Smart For Two bearing with low load capacity), which is generated by three different ways representing the three load cases:

**Load Case 1** with the use of only one hydraulic cylinder generating $F_{k2}$, resulting in 7.2 kN of compressive axial force and 4500 Nm of bending moment acting on the bearing.

**Load Case 2** with the use of both hydraulic cylinders generating $F_{k1}$ and $F_{k2}$, each applying a force up to 3.6 kN resulting in 4500 Nm of pure bending moment acting on the bearing.

**Load Case 3** with the use of only one hydraulic cylinder generating $F_{k1}$, resulting in 7.2 kN of tensile axial force and 4500 Nm of bending moment acting on the bearing. Figure 22 shows measurements for load case 2, which results in the highest deformations from all three scenarios.

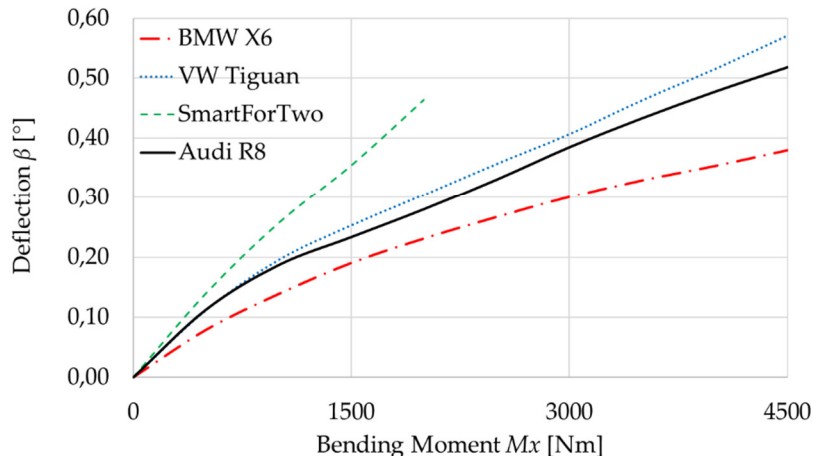

**Figure 22.** Bearing deflection measurement results for different hub bearings, original in specified vehicles and pure bending load case [1].

## 7. Validation of IWM Stiffness

In the 1980s, the need arose for testing the wheel related components in a fast, repetitive and safe way. Mechanical testing systems were built, with which the cumulative damage over the whole lifetime could be reliably reproduced within a few hours [29]. These machines were designed at Fraunhofer Institute for Structural Durability and System Reliability LBF for testing tires, rims and bearings. IWM is a novel component that can greatly benefit from such intensive testing. In our case the Wheel Accelerated Life Test machine (W/ALT) from Fraunhofer LBF is used to verify bearing durability, housing elastic deformation and the air-gap deformation during operation (Figure 23). It generates a realistic wheel contact and side force conditions using a six-axis set of hydraulic cylinders.

The radial electromagnetic forces in the IWM resist the deflection bending, which can be described with an additional moment of force. This counter moment $M_{X\text{-Mag}}$ can be integrated over all magnets, presuming a pure bending of the rotor by:

$$M_{X-\text{Mag}}(\beta) = \sum_{i=1}^{i=N} F_{Y_i} Z_i \sin(\alpha) \tag{10}$$

with $\beta$ being the angular deflection, $N$ is the number of magnets, $F_{Yi}$ is the axial force of i-th magnet, and $Z_i$ is the vertical position of i-th magnet. After integration the results show that this counter moment of force is of the order of magnitude of 1% of the bending moment resulting from the road; thus, the stiffness contribution from magnets is small and can be neglected.

The test procedure was defined as follows: first, the IWM stiffness was measured (characterization sequence), then the highest positive and negative bending moments defined in the requirements were tested five times, alternating in the cornering sequence. The third sequence tested bearing durability and was based on the SAE international J328/2005-2 standard [30]. The load on the bearing was increased by 250% and the hence duration of the accelerated lifetime test could be reduced to 26 hours. The characterization and cornering sequences were repeated after the fatigue tests each time.

During the entire procedure, thermal sensors were placed on the bearing hub and a thermal camera filmed the test. Moreover, two laser sensors measured axial and radial deformation of the rotor.

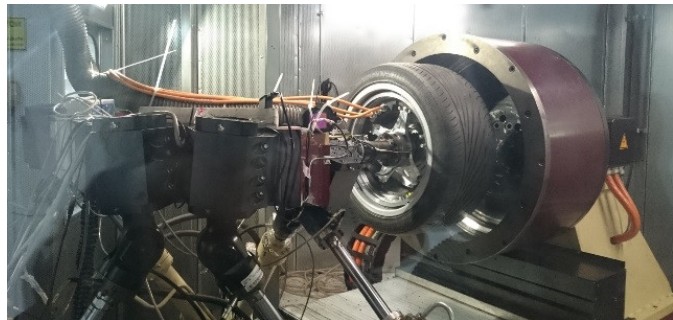

**Figure 23.** IWM in Wheel Accelerated Life test (WALT) for endurance testing at Fraunhofer LBF [1].

The stiffness of the motor is derived from rotor displacement measurements based on the applied loads. The resulting stiffness is not constant but increases with the bending moment. When extremely loaded, the bending stiffness of the bearing can be twice as high as when not loaded. This is due to the load-dependency of the stiffness matrix elements as discussed earlier.

The angular deflection of the rotor is plotted in Figure 24. It can be seen that the difference between before and after the endurance sequence is within 8%.

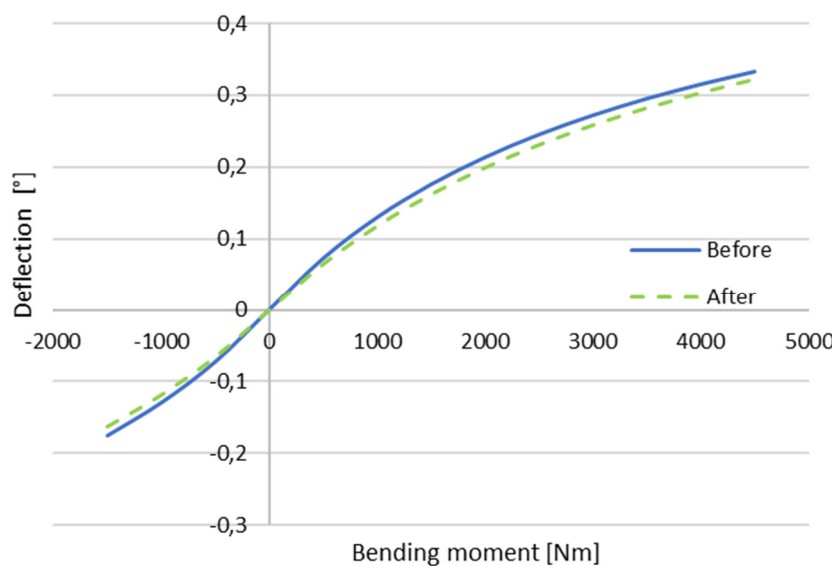

**Figure 24.** Measurements of the deflection for analyzed IWM with BMW X6 hub bearing before and after the endurance sequence [1].

During the test no components were damaged, neither did a contact in the air gap occur. Thermal sensors showed that the bearing never started to overheat. The repetition of the five highest bending moments is visible on the temperature measurements of the bearing (see Figure 25). During the long endurance sequence test, the temperature is raised until equilibrium, which is reached in 15

minutes. The motor's temperature is shown at 0, 5 and 15 minutes in Figure 26. Afterwards, the temperature of the hub bearing remained constant, which indicates that no damage has occurred that could increase the residual torque.

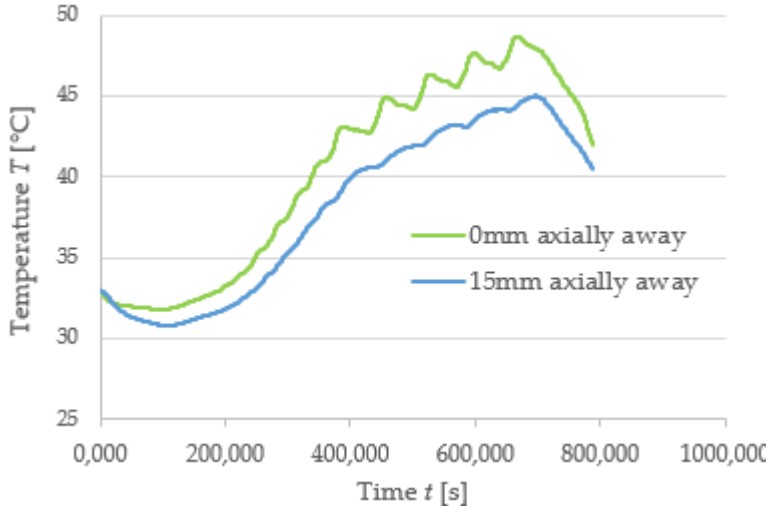

**Figure 25.** Temperature measurements with timestep 1 ms on the hub bearing, directly above ball (green) and 15 mm away (blue) during the first characterization and cornering sequences as shown on Figure 8.

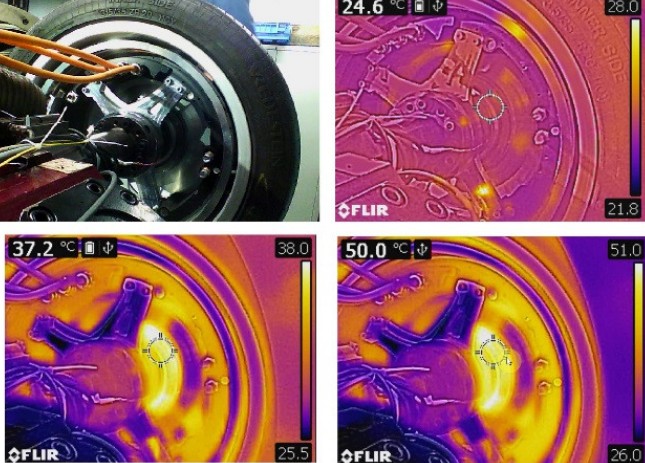

**Figure 26.** Thermal pictures of the motor when the radial endurance sequence starts (from 0 to 15 min) [1].

## 8. Results

The performed tests successfully demonstrate that the bearing can sustain the load and prevent contact in the air gap, which was the main objective. Additionally, bearing deflection tests before and after fatigue tests show that fatigue tests did not result in hub bearing damage, which is also vital for the definition of the product lifetime and maintenance interval. Figure 27 shows high correlation of the deflection tests made on IWM, hub bearing and the predicted characteristics of the analytical bearing design/analysis tool.

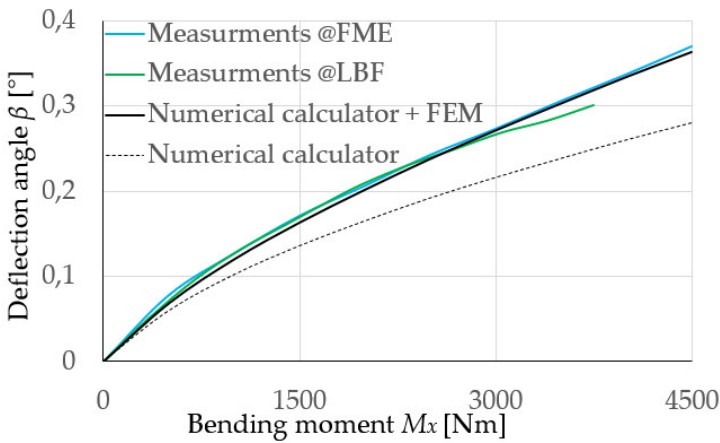

**Figure 27.** BMW X6 bearing deflection comparison.

The higher calculated bearing stiffness compared to the measured one can be explained by the elasticity of hub bearing housing components and also IWM stator and rotor plates, bolt connections as well as the adapters used for fixating the bearing during the deflection tests.

## 9. Conclusions

From the design perspective, the approach presented here drastically simplifies the selection of a suitable hub bearing geometry for a specific in-wheel motor application with the developed analytical bearing design/analysis tool. The hub bearing, being a critical component for in-wheel motors, has more demanding requirements than wheels in conventional propulsion architectures, since the deflection angle affects the motor functionality and output torque. The proposed approach and developed internal EBECA tool allow engineers to design a functional IWM assembly faster, predict the operating turning moment (torque) envelope within a specified application related to driving regime, and defines testing procedure with fatigue lifetime tests. In addition to the hub bearing importance, it must be mentioned at this point that the housing design also has a significant impact on the IWM stiffness, as comprehensively covered in [31].

## 10. Patents

1. WO2012138303A2; Electromagnetic design: Compact multiphase wave winding of a high specific torque electric machine
2. WO2018124971A1; 23465; Electromagentic design of in-wheel motors: Arrangement for determining maximum allowable torque.
3. CT/EP2017/081085; WO/2012/138303/A2; Electric machine with a cooling system and a method for cooling an electric machine
4. SI23465; WO/2013/180663; Electrical gear for electric vehicles with direct drive
5. SI23406; Electric machine with reduced holding torque, with torque vibration and unchanged torque constant
6. PCT/EP2017/079793; WO/2018/095868; Integrated electric gear and charger system for battery powered electric vehicles
7. EP3340439; USA: 20180183292 and EPO: 3340439; Voltage balanced winding pattern for an electric machine with a minimal number of connections and method for assembly of such winding
8. PCT/SI2016/000030; WO/2018/124971; Arrangement for determining maximum allowable torque
9. WO/2019/098949; Method and apparatus for compact insertion of multiphase pseudo helical wave winding into electrical machine
10. WO/2019/139545; In-wheel electric motor maintenance integration
11. WO/2019/151956; Integrated gap retention element for electric motor
12. WO/2020/00966; Electric vehicle energy balance crediting and debiting system and a method thereof

**Author Contributions:** M.B. coordinated the activity, in charge of research background, definition of required tests, correlated activities on electromagnetic responses and functionality from the durability point of view. R.C. identification of acting loads on the in-wheel motor component and definition of load cases and execution of validation test at Fraunhofer LBF. A.G. development and validation of stiffness matrix formulation for double row angular contact ball bearings. S.O. development of bearing stiffness calculator and calculation of stiffness matrices. Definition of Definition and validation tests. S.Z. PI and experimental tests and validation of hub bearing stiffness matrix. R.K. experimental measurements and validation of hub bearing stiffness matrix. All authors have read and agreed to the published version of the manuscript.

**Funding:** The work presented in this article was partially funded by the Slovenian Research Agency as part of the "Modelling in technics and medicine" (code P2-0109) research programme.

**Conflicts of Interest:** The authors declare no conflict of interest.

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
