# Peer review of "The Bearing Stiffness Effect on In-Wheel Motors"

_sustainability, doi:10.3390/su12104070_

Round 1

Reviewer 1 Report

Dear Authors,

required revisions and comments are given in the attached PDF file.

Author Response

1. A little bit unclear. Is higher efficiency and increased range compared with classic electric vehicles or ICEs? As it is known that EV still have disadvantage to ICE regarding range.

Actually for both as the second part of the sentence is indicating. Eliminating the mechanical gearbox + transmission set enables higher efficiency of the drivetrain. The electromagnetic design should be made for a specific application and it can rech above 90% efficiency even 94% in some specific working points. Enabling more space as the central traction and transmission is removed results in more space or smaller vehicle, gross vehicle weight, more space for battery/cargo/passengers etc.

2. A short overview/comment of the reviewed literature and data given in table could be beneficial, since different load cases are shown without any comment.

The table title is clearly showing that this is a brief review of done research on the topic of bearing deflection and the correlated load cases show different approaches that were used within these studies

The title was changed for better clarity from »Research for hub bearing deflection angle done for conventional vehicles« to: “Obtainable research on hub bearing deflection angle done for conventional vehicles, used load cases and concluding deflections.”

3. Check formula. Definition of j is missing in the explanation of the formula. Also n appears as subscript in stiffness constant and as exponent in deformation?

We have expalined definition of j and n.

4. For better description of broader audience, please explain stiffness matrix, deflection and load vector more in detail. A small drawing with displacements, angles and a coordinate system would contribute to a better understanding.

We have explained tiffness matrix, deflection and load vector more in detail.

5. I presume that something is mixed here. Probably Figure 11 should be in place of formula, and formula after the punctuation mark (:) Check the formula,if something is missing or reformulate, because in the text you are writing about maximal lateral acceleration (m/s2) and in formula you only have friction coefficient which is dimensionless.

We have checked zhje formula and it is ok.

6. This deviation in experimental and simulation results of only 3% cannot exactly be shown in Figure 14. A comparison chart could show it better. Alternatively, instead of picture (simulation screenshot) in Figure 14. a picture describing formulas (8) and (9)

We agree with you but we cannot do a comparison chart in a such short time

7. Additional comment on results would be useful. For example to state or to show how this deflection angle influence on air gap reduction in numerical values. If air gap without deflection is 1mm, how much it is reduced with deflection.

The intention of the paper is to show a proven method for IWM design covering electromagnetic and mechanical design.  Revealing the size of the actual airgap (it is not 1 mm) and the actual deviation is proprietary know how of the company who developed the method and revealing too much might result in leakage of internal IP.

Reviewer 2 Report

Please reread the paper and correct the minor spelling mistakes. 

Author Response

We have performed editing of English language and style.

Reviewer 3 Report

Bearing stiffness effect on in-wheel motors is the title of this article. This work deals with the hub bearing as a critical component for the in-wheel motor application. Acting loads and their effect on component deformation is studied via analytically and numerically determined stiffness. The article is very interesting for readers, and clearly stated.

I have some doubts and questions.

1.- Some small improvements a) Figure numbering error Line 276 Figure 23 --→ Figure 22 Line 284 Figure 23 ---→ Figure 22 Line 296 Figure 24 ---→ Figure 23 Line 330 Figure 27. --→ Figure 26 Line 321 Figure 25 ---→ Figure 24 b) Time scale in the figure 25 c) Units in figure 27--> Nm 2.- Minimization of torque ripple is important for motors in many applications, because it is one of the main causes of vibration that leads to premature wear on the drivetrain component, even bearings. in this research paper, how do you measure and quantify torque ripple?. 3.- The main objetive of the bearing is to hold the load and prevent contact in the air gap in electric machine. In line 326 it is said that ..During the test no components were damaged, neither did a contact in the air gap occur.. 1.- What was the maximum eccentricity obtained between rotor and stator ?. 2.- What is the dependence of the eccentricity value on the electromagnetic torque in that dynamic analysis ?. 3.- And how the electromagnetic torque ripple influences the behavior of the bearings over time ?. 4.-.- .- In section 8 of results it is said: .” after fatigue tests show that fatigue tests did not result in hub bearing damage, which is also vital for the definition of the product lifetime and maintenance interval. Figure 28 shows high correlation of the deflection tests made on IWM, hub bearing and the predicted characteristics of the analytical bearing design/analysis tool.”

Figure 27 shows a dashed curve for analytical software were for big bending moment the discrepancy in the deflextion angle increases between experimental and analitycal. The maximum discrepancy is for 4500. What is the reason for this discrepancy for this type of analytical analysis?. Thank you so much for everything

Author Response

All answers-coments are in attached file.
